# Investigating the Friction Behavior of Turn-Milled High Friction Surface Microstructures under Different Tribological Influence Factors

Jonathan Schanner [1,*,†] , Roman Funke [2] , Andreas Schubert [2] and Alexander Hasse [1]

1 Machine Elements and Product Development, Chemnitz University of Technology, 09126 Chemnitz, Germany
2 Micromanufacturing Technology, Chemnitz University of Technology, 09126 Chemnitz, Germany
* Correspondence: jonathan.schanner@mb.tu-chemnitz.de; Tel.: +49-371-531-36828
† Current address: Chemnitz University of Technology, Reichenhainer Straße 70, 09126 Chemnitz, Germany.

**Abstract:** The coefficient of friction (COF) is an important parameter for mechanical engineers to consider when designing frictional connections. Previous work has shown that a surface microstructuring of the harder friction partner leads to a significant increase in the COF. However, the impact of the changes in the tribological system on the COF are not known in detail. In this study, the tribological influence factors such as the nominal surface pressure, the material pairing, lubrication, and the surface properties of the counterbody are investigated. Microstructuring is applied by turn-milling of an annular contact surface of cylindrical specimens. A torsional test bench is used to measure the torque depending on the displacement of the two specimens, thus enabling the determination of the COF. All tests with the microstructured specimens result in higher COF than the reference test with unstructured samples. The manufacturing process of the counterbody surface, the nominal surface pressure, and the materials in contact have a significant influence on the COF. While lubrication reduces friction in the case of unstructured specimens, the COF does not change significantly for microstructured samples. This proves that the deformative friction component dominates over the adhesive. Microstructuring the harder friction partner increases the transmittable torque in frictional connections and reduces the sensitivity towards possible contamination with lubricants.

**Keywords:** coefficient of friction; friction enhancement; microstructuring; static friction; turn-milling; tribology





## 1. Introduction

Frictional connections are widely used to transfer force or torque between two or more components. Companies often redesign and optimize their machine parts and thus the frictional connections in order to reduce manufacturing costs, save resources, and increase efficiency. According to Coulomb's law of friction, the frictional force can be determined as the product of a normal force and the coefficient of friction (COF) [1].

Having the contact pressure at its maximum, the COF remains the only adjustment parameter to increase the frictional forces between the components. For this reason, methods that enable an increase in the COF are of particular interest in research and application.

It is important to understand that the COF is a system property. Several mechanisms occur simultaneously on certain scales, including the surface asperity interaction, molecular forces, and the shear properties of the solids and of the substances between the surfaces [2,3]. Previous studies have shown that three mechanisms of friction can be isolated to describe the effects within a tribological system: adhesion, tribochemistry, and deformation [4–7]. They are influenced by the material pairing, nominal surface pressure, velocity, direction of the transmitted forces, lubrication, and surface properties [6,8–11]. All these factors can be gathered under the term tribological influence factors. Since the COF in this study

is determined in a quasistatic setting, the tribochemical effects can be largely neglected here, as these are only relevant for dynamically loaded systems in which chemical surface changes through adsorption or oxidation occur [6,12]. As described in [4,11,13], it is useful to consider the COF in a static case as the sum of two components: $\mu = \mu_{\text{def}} + \mu_{\text{adh}}$. Hereby, $\mu_{\text{def}}$ includes the deformation mechanisms such as plowing or plastic deformation of asperities, whereas $\mu_{\text{adh}}$ corresponds to the adhesion component of the friction consisting of the molecular bonds, surface energy, and chemical composition of the materials in contact [8,11,14].

There are two ways to modify the initial COF: an increase in the adhesive friction properties or an increase in the deformative friction component. Due to the inherent dependence of the two friction components on each other, an increase in one individual component is rarely possible without affecting the other. However, if systematic changes in the tribological system cause one of the two friction components to increase significantly compared to the other, there is a possibility of an increase in the total COF.

For adhesion, the materials that are in contact represent the greatest influencing factor on the COF. Two identical materials in contact tend to have a higher adhesive friction than two different types of materials [15]. Adhesion is also affected by the real contact area between the two surfaces and the lubrication of the system [8]. In most cases, changing the materials is not the most straightforward way to increase the COF, as component strength, stiffness, and weight often have to be maintained. Furthermore, changing the COF by means of adhesion is susceptible to surface contamination [16]. Therefore, it is more effective to change the COF in terms of the deformation component.

In frictional connections, a higher surface roughness tends to increase the friction due to a higher deformation component resulting from the plowing or interlocking of asperities, respectively [17,18]. While changing the surface roughness is an easy solution, the impact is not as large as with other methods, as shown in [19,20]. Another way to increase the deformation is to include particles in contact. These particles need to be harder than both materials to create a micro contact, which lead to a greater resistance against external forces [21]. Most commonly, superhard materials such as diamonds or ceramics are used. These particles can either be directly applied to the part as a coating or can be used as a separate component embedded in a nickel-composite matrix [21,22]. A similar approach is taken through laser microstructuring. The laser beam is used to locally change the geometrical and physical properties of the workpiece surface. Due to the induced heat, the material changes its structure leading to an increase in hardness [23–25]. Similar to the hard particles, the achieved protruding microstructures penetrate the surface of the counterpart and thus create interlocking structures between the two surfaces to increase the COF [25]. While this process eliminates the need for additional parts, it introduces another step in the finishing process of the parts and therefore leads to higher manufacturing costs.

While particles and laserstructuring increase the COF by changing the interlocking and indentation of the contacting surfaces, both processes also increase the costs of the final product by changing the surface finishing process or by introducing a third part into the tribological system. In order to eliminate these additional costs and avoid the need for further components, friction-enhancing microstructures can be created mechanically based on the findings from [17–19]. Here, the microstructures are generated as part of the final machining step of the friction partners.

In [26], turn-milling was used to microstructure specimens of the steel 1.7225 in quenched and tempered heat-treatment conditions (+QT). COFs up to $\mu_{\text{max}} = 0.69$ were achieved in a standard torsional friction test bench at a nominal surface pressure $p_{\text{nom}} = 100\,\text{MPa}$ using fine-turned counterbodies made of the steel 1.0503. This demonstrates a promising increase in COF due to microstructuring. Comparisons with reference tests with face-turned surfaces without additional microstructures showed not only an increase in the COF but also a completely different frictional behaviour due to the microstructures [27]. The measured sliding curves of the microstructured specimens exhibited a distinct local maximum, which indicated a high deformative friction component. Further investigations revealed that

the COF tended to increase with a growing nominal surface pressure between 100 MPa and 300 MPa. The mean values ranged between $\mu_{\text{max}} = 0.45$ at $p_{\text{nom}} = 100$ MPa, and $\mu_{\text{max}} = 0.54$ at $p_{\text{nom}} = 300$ MPa [28]. By changing the tool inclination angle, an asymmetrical profile of the microstructures in the load direction was created, which resulted in direction-dependent friction properties [28].

So far, the friction-enhancing effects of turn-milled microstructures have only been investigated on one material pairing under ideal tribological conditions. In most cases, however, there are no ideal conditions present when using frictional connections. Undesirable environmental influences such as lubricants can strongly influence the friction behavior between two or more components in contact. However, it has not yet been clarified how well the microstructures perform under such changes. In order to be able to better classify the applicability of the microstructures, the tribological influence factors must be investigated in detail.

In this study, a comprehensive investigation is performed to determine the dependence between the COF and tribological influences, which include changes in the material pairing and the properties of the counterbody surface, as well as the nominal surface pressure $p_{\text{nom}}$, for one specific microstructure variant of turn-milled surfaces.

Another essential goal of the investigations is to determine whether the effectiveness of the microstructures is actually due to an increase in the deformation component of the friction $\mu_{\text{def}}$. The use of lubricants during the bench tests reduces the adhesion component $\mu_{\text{adh}}$ to a minimum and therefore allows for an individual determination of each friction component. This assumption is only true for the boundary lubrication regime on the Stribeck curve. It is assumed that a strong influence of the microstructures minimizes the effect of the boundary lubrication.

## 2. Materials and Methods

### 2.1. Evaluation of the Coefficient of Static Friction

To determine the COF, a test bench of IKAT at Chemnitz University of Technology was used (Figure 1). Cylindrical specimens with a clamping diameter of 45 mm and a length of 65 mm were used. The contact area was an annular surface with an inner diameter of $D_{\text{i}} = 15$ mm and an outer diameter of $D_{\text{o}} = 30$ mm. During the tests, two specimens were pressed together coaxially with a predefined normal force $F_{\text{N}}$. Due to the specified contact area, the resulting nominal surface pressure $p_{\text{nom}}$ was derived. The specimens were then rotated against each other by a defined angle $\varphi$ using the hydraulic torque actuator. Due to the friction between the specimens, shear stresses were developed in the contact. The resulting torque $T_{\text{R}}$ was measured with strain gauges as a function of the relative displacement in the contact. The following equation was used to define the friction diameter $D_{\text{m}}$ for an annular contact surface with an inner and outer diameter $D_{\text{i;o}}$:

$$D_{\text{m}} = \frac{2}{3}\left(\frac{D_{\text{o}}^3 - D_{\text{i}}^3}{D_{\text{o}}^2 - D_{\text{i}}^2}\right). \tag{1}$$

For the given specimen geometry, a friction diameter of $D_{\text{m}} = 23.3$ mm was determined. Using the friction diameter $D_{\text{m}}$, the sliding distance $s_{\text{R}}$ between the two specimens was derived using the following equation:

$$s_{\text{R}} = \varphi \cdot \frac{D_{\text{m}}}{2}. \tag{2}$$

An angular displacement of $\varphi = 5°$ with a constant rotational speed of $\omega = 0.5\,°\,\text{s}^{-1}$ was set to determine the COF under quasistatic conditions. This angle corresponded to a sliding distance of $s_{\text{R}} \approx 1000\,\mu\text{m}$ along $D_{\text{m}}$ between the two specimens. Because the COF between components of different sizes may be evaluated, the use of displacement as a distance $s_{\text{R}}$ is more suitable than the description by angle $\varphi$. This allows for a better comparison between real parts and the model tests [29]. To determine the COF $\mu$ at the

friction diameter $D_m$ for a given torque $T_R$ and the normal force $F_N$ the following Equation was used:

$$\mu = \frac{2 \cdot T_R}{D_m \cdot F_N}.$$

(3)

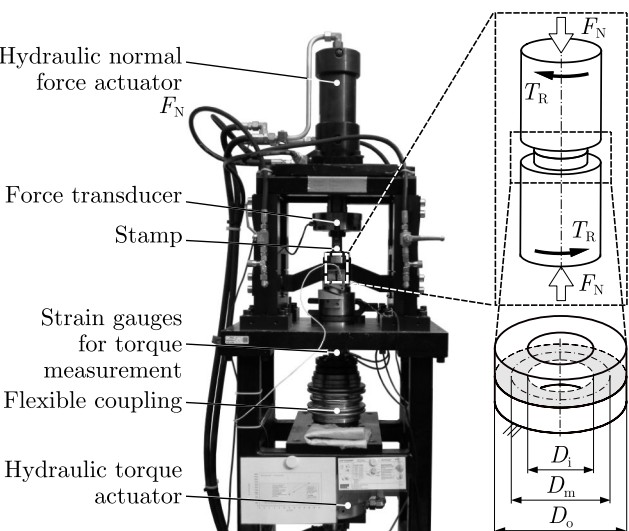

**Figure 1.** Friction test bench (**left**) and specimens with load principle (**right**).

The measured torque was then plotted against the displacement to determine the representative COF, following the test methods described in [30–32]. This procedure was necessary because of the different possible friction characteristics that can occur during testing [30,31,33]. The different characteristics (Figure 2) can hint toward the distribution of the friction in terms of the adhesion and deformation. *Characteristic A* indicates a dominating deformative friction component because of the shearing mechanisms of the roughness asperities, while the *B and C characteristics* often indicate a higher adhesive friction. A friction *characteristic B* is defined by an abrupt transition between sticking and sliding, while *characteristic C* is indicated by a smoother transitioning section. An example for the evaluation of the representative COFs is given in Figure 2 [32]. The indices for $\mu$ describe the irreversible displacement in the contact, determined by subtracting part of the displacement due to the elasticity of the system (test-bench, contacting material elasticity). Hereby, $\mu_{20}$ describes the COF where $s_R = 20\,\mu\text{m}$ irreversible slip occurred between the two specimens. In addition to the displacement dependency of $\mu_i$, $\mu_{max}$ describes the maximum COF, which occurs locally for *A characteristics*. The corresponding displacement $s_{R,max}$ is defined as a variable depending on the position of $\mu_{max}$ [30,32]. Depending on the resulting friction characteristics, either $\mu_{20}$ or $\mu_{max}$ are evaluated as representative COF for the tested specimens.

This methodology was carried out for all test variations. Afterwards, the mean value $\bar{x}(\mu)$ and standard deviation $\sigma(\mu)$ of the COF for the five tests were determined.

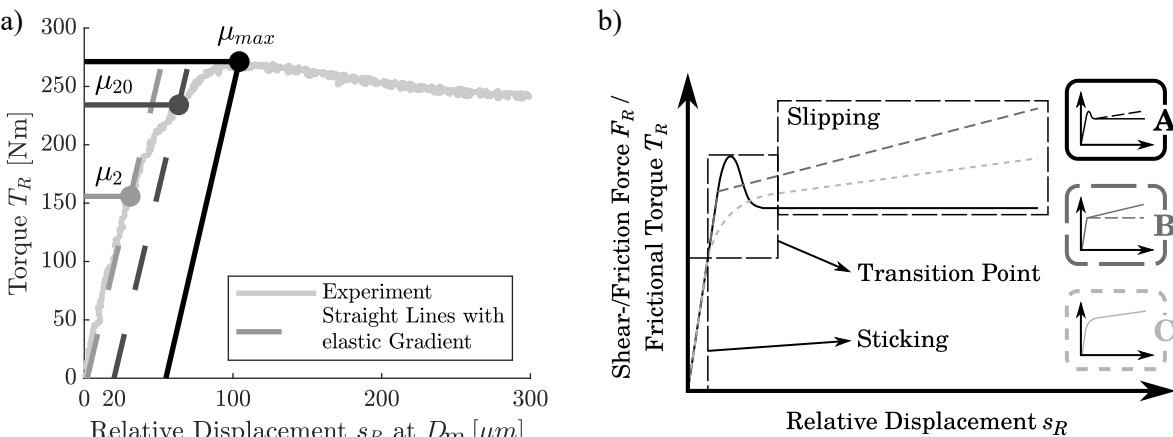

**Figure 2.** Evaluation of the COF: method to determine the representative COF using sliding curves (**a**) and different possible friction characteristics according to [30,31] (**b**).

### 2.2. Surface Microstructuring

Turn-milling was used for microstructuring, applying the kinematics shown in Figure 3. The spiral tool path resulted from the superposition of two feed components: the workpiece rotation and the translatory movement of the tool towards the workpiece axis. The cutting movement was determined solely by the rotation of the tool. The radial feed $f_{rad}$ corresponded to the distance covered by the tool during one workpiece revolution. The feed per tooth $f_z$ resulted from the rotation of the end mill. The height of the profile $Rt_f$ as well as the profile tip angle $\alpha_{pt}$ were determined by the corner geometry, the tool inclination angle $\beta$, and $f_z$.

The process parameters and the tool geometry were selected on the basis of previous investigations [27]. The tools used were TiAlN coated single-edged cemented carbide end milling cutters with a diameter of 6 mm and a sharp corner with a tool-included angle of $\varepsilon_r = 88°$. To achieve a symmetric profile, the tool inclination angle was set to $\beta = 46°$. The depth of cut and the cutting speed were kept constant at $a_p = 0.2$ mm and $v_c = 100$ m / min$^2$, respectively. A radial feed $f_{rad} = 0.2$ mm and a feed per tooth $f_z = 0.125$ mm were applied. All experiments were conducted on a milling center KERN Pyramid Nano, using a minimum quantity of lubrication (Figure 3b).

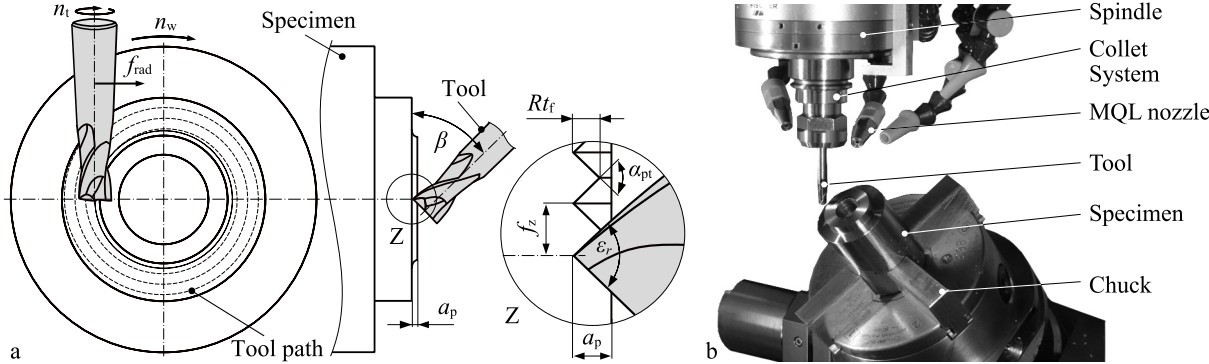

**Figure 3.** Turn-milling: (**a**) kinematics and (**b**) machine room.

### 2.3. Surface Evaluation

The surface evaluation was conducted with a 3D laser scanning microscope, Keyence VK-X1000, with a measurement accuracy of 0.2 μm for the used setup. For the microstructured surfaces, a measuring field with a size of 0.5 mm × 6 mm was used. In contrast to the standard DIN EN ISO 25178, it thus deviated from a square shape. However, this adjustment was made to cover the inhomogeneous distribution of the microstructures in

the radial direction (cf. [27]). The unstructured surfaces of the counterbodies were analyzed within a 3 mm × 2.5 mm section of the contact surface. The filtering of the roughness was performed in accordance with ISO 11562 using a cutoff wavelength of 0.25 mm. Figure 4 shows the positioning of the measured sections on the annular surface of the specimen.

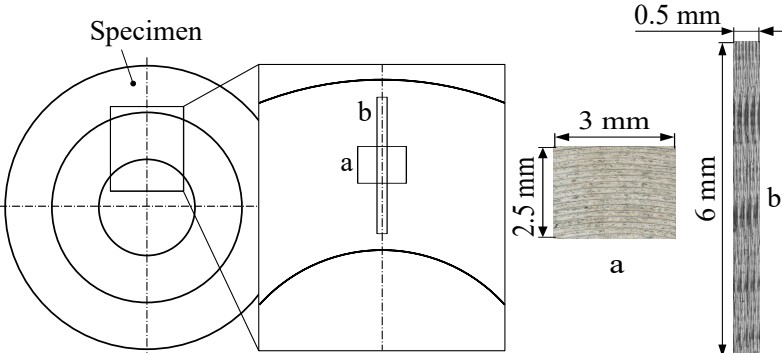

**Figure 4.** Size and position of the measuring field for: (**a**) unstructured and (**b**) microstructured specimens.

The surfaces of all the microstructured specimens and selected counterbodies were measured before and after testing. The analysis of the surfaces focused on estimating the geometry of the microstructures and the occurring plastic deformations of the microstructured and counterbody surfaces. Due to the structural change in the radial direction, an analysis by line roughness was not useful. Therefore, the surface texture parameters were used. Here, a distinction was made between the different parameter groups. According to DIN EN ISO 25178 a separation into height, space, hybrid, functional, and functional volume parameters is possible [34]. For microstructures, the structure size and volume are of particular interest, which is why the height, hybrid, and functional volume parameters were used for evaluation [34,35]. A more detailed explanation of the specific parameters used in this study is given in Section 3.1. In order to establish the associations between the surface properties and the COF, each surface parameter was correlated with the measured COF for each test variation. Since these were normally distributed metric parameters, the Pearson correlation coefficient $r^2$ was used to estimate the magnitude of the relations.

### 2.4. Tribological Influence Factors

To determine the influence of the materials in contact, two industrially relevant material pairings were investigated: 42CrMo4 (1.7225) in quenched and tempered heat treatment conditions (+QT) vs. C45 (1.0503) and C45 vs. EN-GJS-400 (0.7040). These material combinations also represented two different types of connections, one steel vs. steel and one steel vs. cast-iron pairing, which are often used as representatives of the respective applications in the industry [36]. The chemical composition of the materials is shown in Table 1. The following hardness values were determined for the different materials: 340 HV10 (42CrMo + QT), 200 HV10 (C45), and 170 HV10 (EN-GJS-400). For each pairing, the harder of the two friction partners was microstructured by turn-milling as described in Section 2.2.

**Table 1.** Chemical composition in percent by weight of the materials.

| Element / Material | C | Si | Mn | P | S | Cr | Mo | Ni | Cu |
|---|---|---|---|---|---|---|---|---|---|
| 42CrMo4 + QT | 0.4 | 0.22 | 0.75 | 0.021 | 0.027 | 1.02 | 0.17 | - | - |
| C45 | 0.42 ... 0.5 | <0.4 | 0.5 ... 0.8 | <0.045 | <0.045 | <0.4 | <0.1 | <0.4 | - |
| GJS-400 | 3.4 ... 3.6 | 2.7 ... 2.9 | <0.1 | <0.04 | <0.012 | - | - | - | <0.1 |

The different parameters of the tribosystem are shown in Table 2. In order to investigate every interaction, tests were carried out for all parameter combinations. In addition to the

parameters listed in Table 2, each combination was tested with paraffin jelly as lubricant to minimize the adhesive friction component. Using the expression $\mu = \mu_{\text{def}} + \mu_{\text{adh}}$, the deformative friction component $\mu_{\text{def}}$ can be measured for the lubricated surfaces, where $\mu_{\text{adh}}$ is minimized. Using the results of the dry and lubricated tests, the adhesive component $\mu_{\text{adh}}$ can be derived as $\mu_{\text{adh}} \approx \mu_{\text{dry}} - \mu_{\text{lubricated}}$. In order to obtain a sufficient accuracy, five tests were carried out for each combination, resulting in $n = 120$ total tests. Specimens with no additional microstructuring were tested at $p_{\text{nom}} = 100$ MPa for each material pairing and served as reference. The samples for these tests were machined with the specification *fine turned*.

**Table 2.** Investigated influence factors.

| Materials in Contact | Manufacturing Process of the Counterbody | Nominal Surface Pressure $p_{\text{nom}}$ |
|---|---|---|
| 42CrMo4 + QT (struc.) vs. C45 | Fine turned | 30 MPa |
| C45 (struc.) vs. EN-GJS-400 | Rough turned | 100 MPa |
| | Ground | |

Three different surface conditions of the counterbodies were investigated: fine turned, rough turned and ground. The resulting surface sections are shown in Figure 5. Rough turning was performed using an indexable insert with a corner radius $r_\epsilon = 0.4$ mm and applying a feed $f = 0.15$ mm. For fine turning, an insert with a wiper geometry of the type CCMT09T304-WS from Sandvik and $f = 0.1$ mm were utilized. Both turning operations were performed at a depth of cut $a_{\text{p}} = 0.3$ mm and a cutting speed $v_{\text{c}} = 200$ m / min. The third surface state was created by horizontal-spindle rotary table surface grinding. This resulted in concentric grinding grooves. Due to the different material properties, the topography and roughness values of the surfaces machined identically differed quite significantly, as shown in Figure 5. The roughness values of the specimens made of cast iron were considerably higher than those of the steel specimens. The fine-turned specimens of 42CrMo4 + QT exhibited a roughness $Rz \approx 1$ μm.

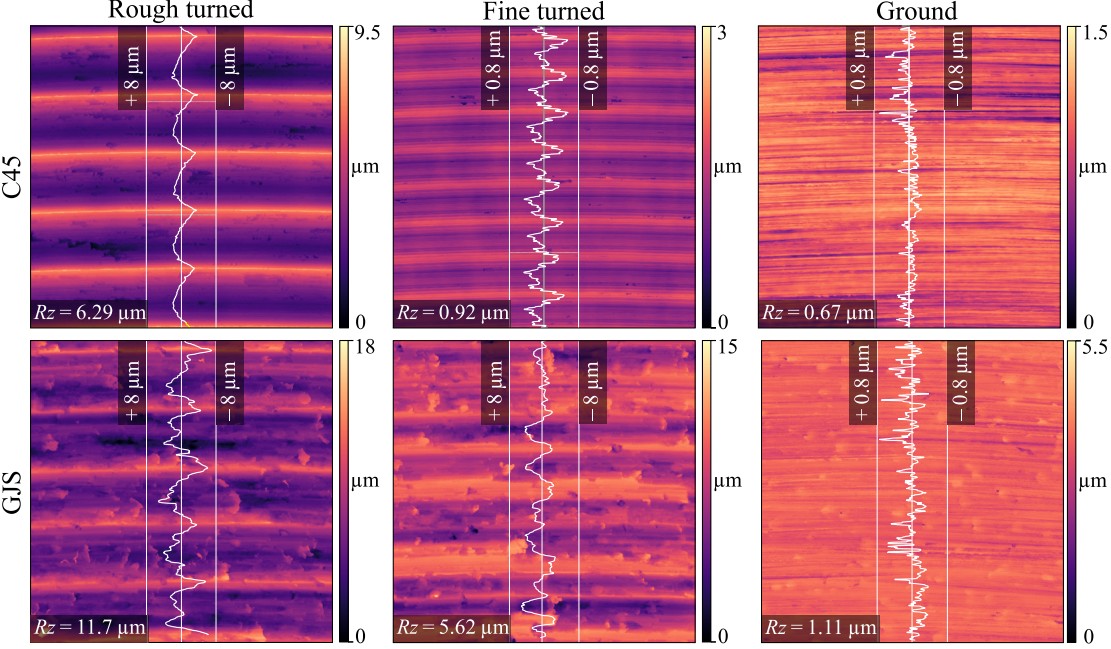

**Figure 5.** Surface sections (0.8 mm × 0.8 mm) of the examined counterparts before testing.

## 3. Results and Discussion

### 3.1. Surface Topography of Microstructured Specimens

First, the surfaces of the microstructured test specimens were measured before the bench tests. Influences of the turn-milling parameters were shown in [27] for one of the materials, 42CrMo4 + QT. Emphasis was placed on the resulting structural geometry, which differed depending on the material. Using C45 as a different material for structuring while keeping the turning-milling parameters the same could result in different microstructures compared to the 42CrMo4 + QT. The resulting surface for both materials structures are shown in Table 3. *Sa* describes the mean height of the surface, and *Sdq* is calculated as a root mean square of the slopes at all points in the definition area. For the functional volume parameters, *Vmp* is the peak material volume. The *Vmp* can be derived from the Abbot Curves by using the areal material ratio values between 0–10%. In contrast, *Vvv* is the dale void volume from 80–100% areal material ratio. No significant differences in the surface parameters were found for either of the two materials. Both surfaces showed a gradient of the structure dimensions in the radial direction of the specimen, leading to smaller structures close to the center of the specimen. In contrast to the unstructured surfaces, where the grooves were aligned coaxially to the load direction, the microstructures were perpendicular to the axis of rotation.

**Table 3.** Surface height profiles and surface structure parameters of selected microstructured specimens before testing.

| Surface Height Profile (µm) | Sa (µm) | Sdq (rad) | Vmp (mL m$^{-2}$) | Vvv (mL m$^{-2}$) |
|---|---|---|---|---|
| 42CrMo4+QT  | $6.95 \pm 1.40$ | $1.38 \pm 0.21$ | $0.74 \pm 0.12$ | $0.37 \pm 0.03$ |
| C45  | $7.17 \pm 0.27$ | $1.30 \pm 0.58$ | $0.82 \pm 0.08$ | $0.35 \pm 0.02$ |

### 3.2. Friction Properties of the Unstructured Specimens

In order to determine the influence of the microstructures, as well as the tribological system, reference tests were carried out with unstructured test specimens. For all variants, a fine-turned specimen was paired with a counterbody in one of the surface conditions described in Table 2. The nominal surface pressure was set to $p_{nom} = 100 \, \text{MPa}$. The results of the lubricated and dry condition for the material pairing C45 vs. EN-GJS-400 are shown in Figure 6. The mean values and the standard deviation as well as the corresponding sliding curves for all tested parameters are shown. It can be seen that with an increasing relative displacement $s_R$, the COF $\mu$ also increased, indicating a *B and C characteristic* of the sliding curves. Because no clear indication of a local maximum was found, the corresponding value of $\mu_{20}$ was selected to determine the representative COF [19]. As expected, the $\mu_{20}$ decreased significantly in the lubricated state from $\approx 0.18$ to $\approx 0.10$. The decrease in the COF for the lubricated surfaces and the fact that the connections were showing *B and C characteristics* indicate a prominent adhesive friction component for this material pairing.

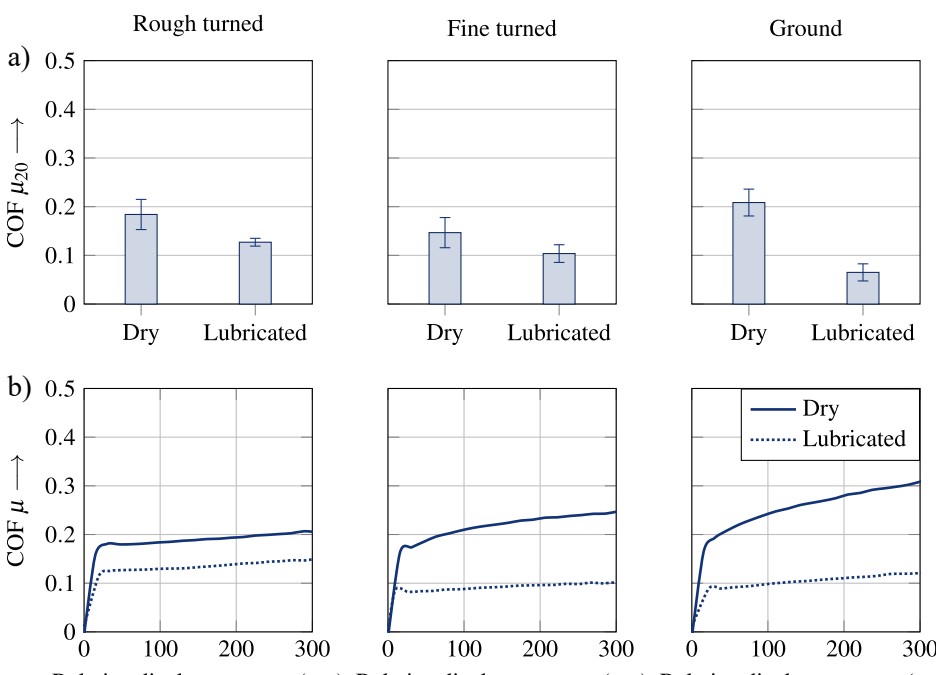

**Figure 6.** Results of the friction tests with the unstructured surfaces of the material pairing C45 vs. EN-GJS-400 at $p_{nom} = 100\,\text{MPa}$; mean values of $\mu_{20}$ (**a**); corresponding sliding curves (**b**).

The results of the tests for the material pairing 42CrMo4 + QT vs. C45 are presented in Figure 7. For this material pairing, all variants showed friction *B and C characteristics* as well. The rising slip curve was particularly noticeable for the fine-turned surfaces. A significant decrease in the COF was seen for the tests with the lubricated surfaces, similar to the material pairing C45 vs. EN-GJS-400. This confirmed the initial assumption that the COF was mainly dependent on the adhesive friction component.

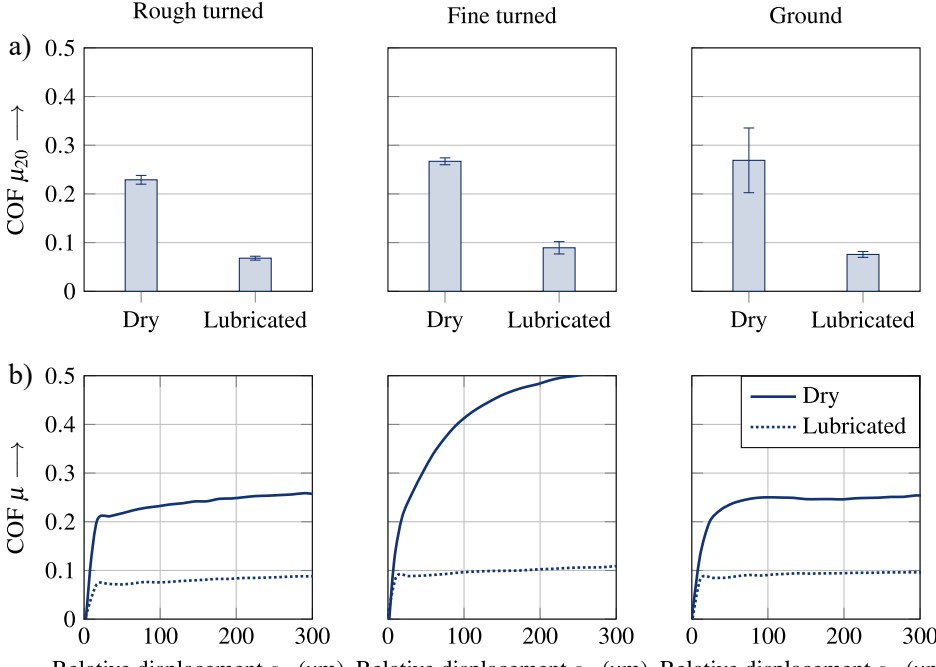

**Figure 7.** Results of the friction tests with the unstructured surfaces of the material pairing 42CrMo4 + QT vs. C45 at $p_{nom} = 100\,\text{MPa}$; mean values for $\mu_{20}$ (**a**); characteristic sliding curves (**b**).

The results of the two material pairings are summarized in Table 4. The unstructured samples showed a clear dependence of the COF on the respective material pairing. The main reason for these differences was seen in the intimate lubrication of the steel vs. cast iron pairing, which occurred due to the embedded graphite particles inside the cast iron [37]. The lubrication effect of graphite occurred after the sliding was initiated, when the damaged graphite particles led to the formation of graphite films, which acted as a solid lubricant [38]. This lubrication, as well as the differences in material hardness, chemical composition, and strength, reduced the adhesive friction component $\mu_{adh}$. The adhesive friction component is mainly responsible for the friction when the direction of the grooves is aligned in the direction of the movement/force [37,39]. This becomes evident when analyzing the surfaces after testing. As shown in Figure 8 microwelding occurred for the fine-turned surfaces of the pairing 42CrMo4 + QT vs. C45, which also shows the highest increase in the COF at greater displacements. This is also the only surface configuration where micro-welding dots are clearly visible under the microscope. The effect is shown in the respective sliding curves, where the fine-turned surfaces of 42CrMO4 + QT vs. C45 reached the highest COF. For both material pairings, the dependence of the adhesion was evident when comparing the results of the lubricated and dry surfaces. The variants with lubricated surfaces reached a significantly lower COF. The lubricant reduced the adhesive component of the friction to a minimum, resulting in a lower overall COF $\mu$. The remaining friction for the lubricated variants was therefore mostly attributed to the deformative friction component $\mu_{def}$, where both material pairings achieved similar COF values. The biggest drop in COF arose for the ground surfaces of 42CrMo4 + QT vs. C45, where the lubrication reduced the representative COF from $\mu_{20,dry} = 0.27$ down to $\mu_{20,lubricated} = 0.08$. A comparison of the COF distribution $\mu = \mu_{def} + \mu_{adh}$ showed that for 42CrMo4 + QT vs. C45, 30% of the friction could be attributed to the deformation and 70% to the adhesion. For C45 vs. EN-GJS-400, the distribution was 60%: 40%. The determined mean values also agreed well with the values in the literature, such as [8,36].

**Table 4.** COF $\mu_{20}$ of $n = 3$ tests for the unstructured surfaces.

| Materials | Surface Structure | Lubrication | $\bar{x}(\mu); \sigma(\mu)$ |
|---|---|---|---|
| C45 vs. EN-GJS-400 | Rough turned | Dry | $0.18 \pm 0.01$ |
| | | Lubricated | $0.13 \pm 0.01$ |
| | Fine turned | Dry | $0.15 \pm 0.03$ |
| | | Lubricated | $0.10 \pm 0.02$ |
| | Ground | Dry | $0.21 \pm 0.03$ |
| | | Lubricated | $0.07 \pm 0.02$ |
| 42CrMo4+QT vs. C45 | Rough turned | Dry | $0.23 \pm 0.02$ |
| | | Lubricated | $0.07 \pm 0.01$ |
| | Fine turned | Dry | $0.27 \pm 0.01$ |
| | | Lubricated | $0.09 \pm 0.01$ |
| | Ground | Dry | $0.27 \pm 0.07$ |
| | | Lubricated | $0.08 \pm 0.01$ |

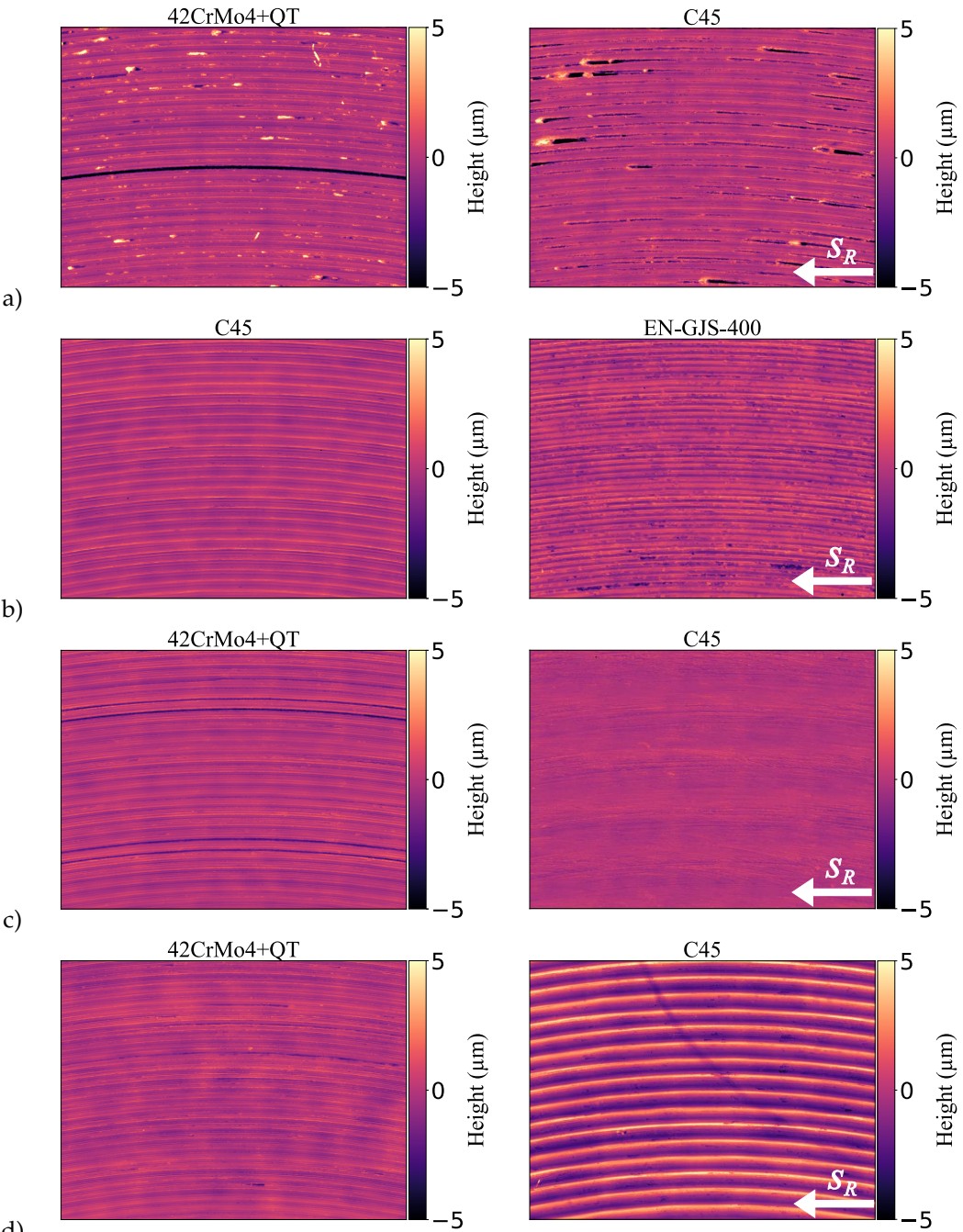

**Figure 8.** Surface height profiles of the reference test surfaces after $s_R = 1000\,\mu m$ displacement under the dry condition. 42CrMo4+QT vs. C45, fine turned (**a**); C45 vs. EN-GJS-400, fine turned (**b**); 42CrMo4+QT vs. C45, ground (**c**); 42CrMo4 + QT vs. C45, rough turned (**d**).

### 3.3. Friction Properties of the Microstructured Specimens

The results for the combination C45 vs. EN-GJS-400 are shown in Figure 9. In contrast to the reference tests, a local maximum of the COF occurred in the range of $s_R = 10\,\mu m$–$100\,\mu m$ for all variants, resulting in a friction *characteristic of type A*. Therefore, the evaluation of the representative COF was changed to $\mu_{max}$, as described in Section 2.1. Compared to the unstructured specimens, there was a clear increase in the representative COF for all variants. A trend can be seen, where $\mu_{max}$ shifted to a higher $s_{R,max}$ in the tests with a higher nominal surface pressure $p_{nom}$. For most variations, the COF increased with an increasing pressure. This may be due to the increased shear resistance of the microstructures, which occurred as a result of

the deeper penetration of the structures into the counterbody. In addition, the lubrication of the test specimens seems to have had less influence on the COF compared to the unstructured specimens, indicating an increase in the deformative friction component for the microstructured specimens compared to unstructured ones. After the initial slipping of the specimens, the later portion of the curves ($s_R > 200\,\mu m$) tended to reach the same COF values for both pressures. This behavior can be explained by the failure of the microstructures.

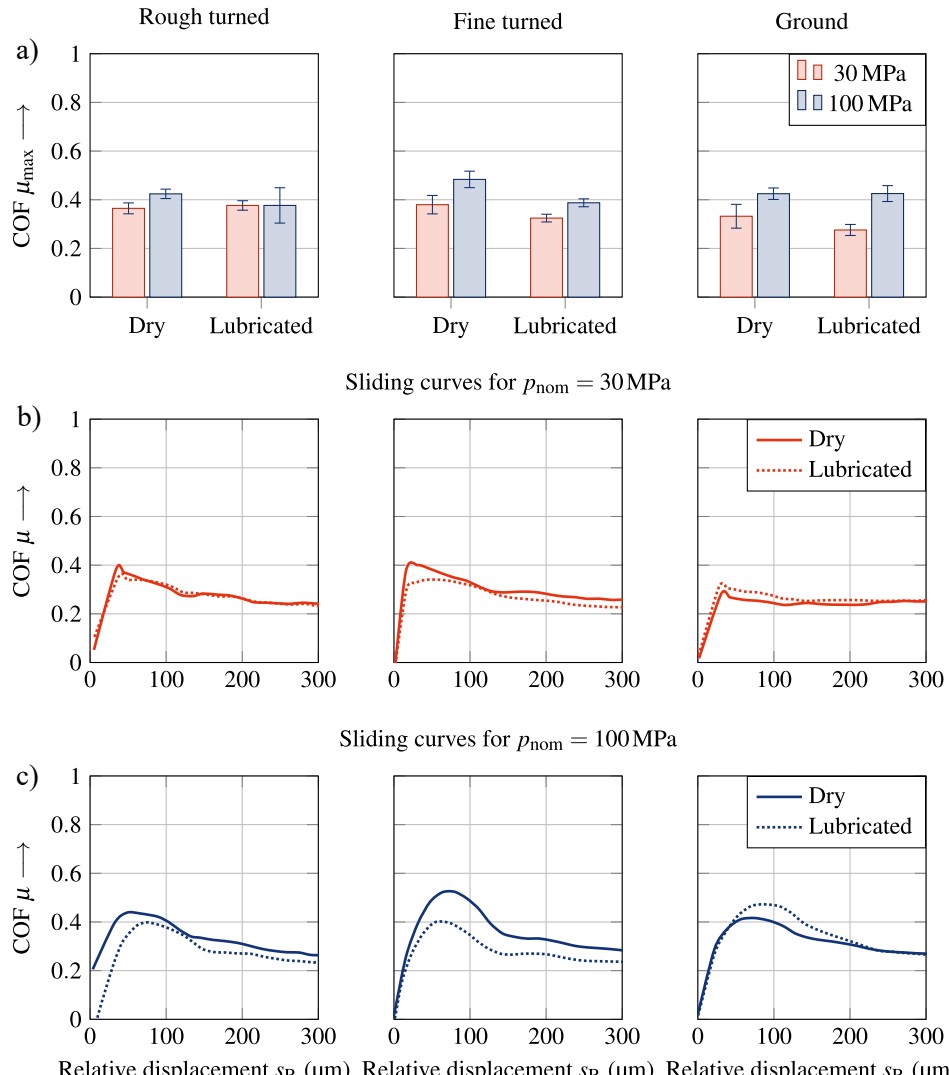

**Figure 9.** Results of the friction tests for the material pairing C45 vs. EN-GJS-400 with the microstructured specimens; mean values for $\mu_{max}$ (**a**); characteristic sliding curves for $p_{nom} = 30\,MPa$ (**b**); characteristic sliding curves for $p_{nom} = 100\,MPa$ (**c**).

The results for the combination 42CrMo4 + QT vs. C45 are shown in Figure 10. Similar to the other material pairing, an increase in the representative COF can be seen for all variants compared to the unstructured specimens. A clear local maximum was found for all test variations. The shifting of $s_{R,max}$ at a higher nominal surface pressure $p_{nom}$ was also present for this material pairing. The increase in the COF at higher pressures $p_{nom}$ was comparable to the material pairing C45 vs. EN-GJS-400. Additionally, no clear influence of the lubrication on the COF was detected, when comparing the lubricated and dry variants. In contrast to the other material pairing, the friction values did not approach the same COFs for both pressures after reaching a sliding distance of $s_R > 200\,\mu m$. For the higher nominal surface pressure $p_{nom} = 100\,MPa$, the difference between the lubricated and dry surfaces increased for larger sliding distances.

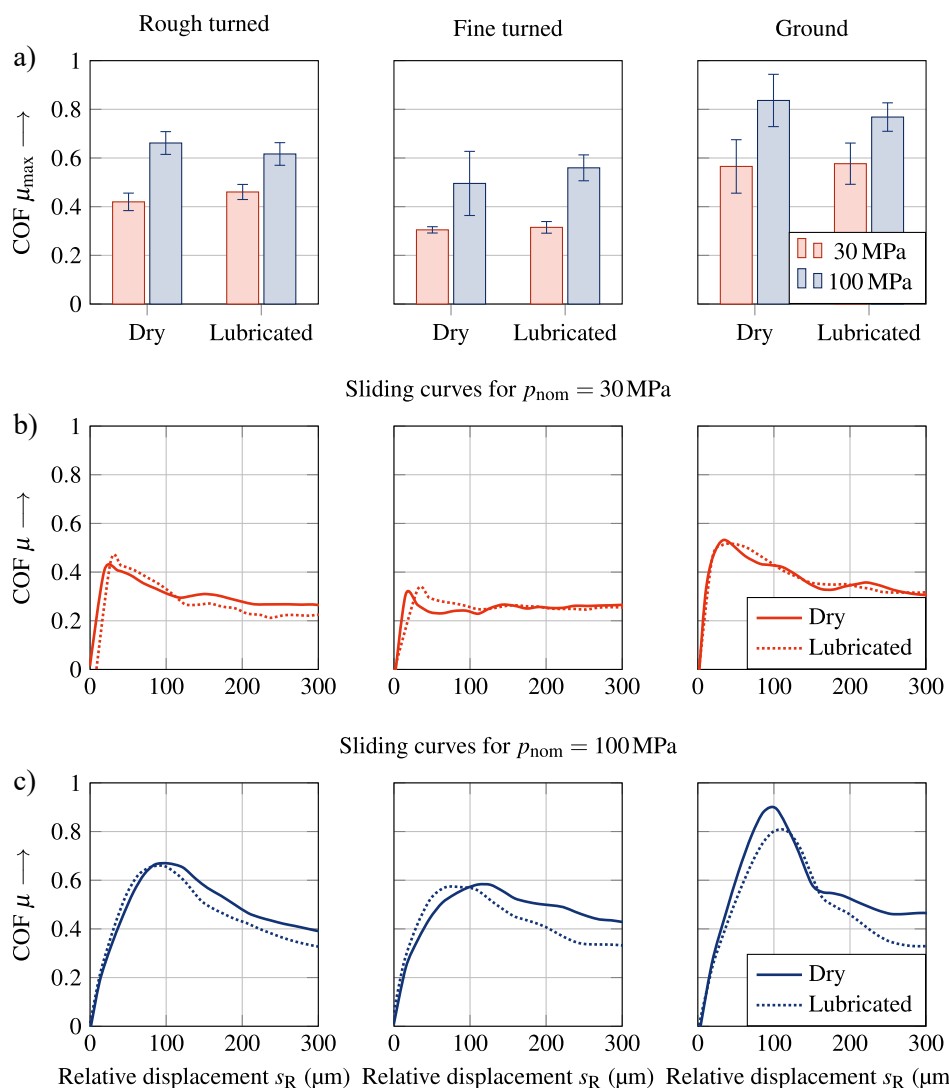

**Figure 10.** Results of the friction tests for the material pairing 42CrMo4 + QT vs. C45 with the microstructured specimens; mean values for $\mu_{max}$ (**a**); characteristic sliding curves for $p_{nom} = 30\,\text{MPa}$ (**b**); characteristic sliding curves for $p_{nom} = 100\,\text{MPa}$ (**c**).

A comparison of the representative COF for all parameter variations is shown in Table 5. As stated earlier, for all the tests, an increase in the representative COF was seen, compared to the unstructured specimens. They also showed a change in the friction characteristic. While *B and C characteristics* occurred for the unstructured variant, the experiments with microstructured specimens showed an *A characteristic* with a pronounced local maximum. This change is an indicator for a shift to a higher deformative friction component, due to the penetration of the microstructures into the counterbody surface. The penetrations led to micro-interlocking between the two specimens. Another factor that increased the friction was the orientation of the microstructures on the specimen surface. While the grooves of the unstructured test specimens were coaxial to the load direction, the microstructures were perpendicular to the rotation axis. This alignment led to increased resistance against the deformation of the microstructures in contact. When comparing the different material pairings, significantly higher values of the COF for ground surfaces were seen for the material pairing 42CrMo4 + QT vs. C45. A maximum mean value of $\bar{x}(\mu_{max}) = 0.84$ was determined for the nonlubricated surfaces. Compared to a value of $\bar{x}(\mu_{max}) = 0.42$ for C45 vs. EN-GJS-400, the COF almost doubled. Higher COFs of the pairing 42CrMo4 + QT vs. C45 were observed for all combinations with a nominal surface pressure $p_{nom} = 100\,\text{MPa}$, as well as most of combinations with $p_{nom} = 30\,\text{MPa}$.

The difference between the two material pairings can be explained due to the higher shear strength of the 42CrMo4 + QT material. Since the geometry of the microstructures was similar for both material pairings, the failure of the microstructure was only dependent on the penetration depth and shear strength of the structured specimen. For the materials at hand, 42CrMo4 + QT showed 1.5 times the shear strength of C45 and thus transmitted significantly higher torques before the microstructures failed. The distribution of the friction components clearly changed towards the deformative friction component. For the microstructured variant of 42CrMo4 + QT vs. C45, the deformative friction component was responsible for 95% of the total friction compared to 30% from the unstructured variants. An increase to 90% deformative friction for C45 vs. EN-GJS-400 confirmed the thesis that the microstructures performed nearly the same under lubricated and dry conditions.

**Table 5.** COF $\mu_{max}$ of $n = 5$ tests with the microstructured surfaces.

| Materials | Counterpart Surface Structure | Lubrication | Pressure $p_{nom}$ | $\bar{x}(\mu); \sigma(\mu)$ |
|---|---|---|---|---|
| C45 (struc.) vs. EN-GJS-400 | Rough turned | Dry | 30 MPa<br>100 MPa | $0.36 \pm 0.02$<br>$0.42 \pm 0.02$ |
| | | Lubricated | 30 MPa<br>100 MPa | $0.38 \pm 0.02$<br>$0.38 \pm 0.07$ |
| | Fine turned | Dry | 30 MPa<br>100 MPa | $0.38 \pm 0.04$<br>$0.48 \pm 0.03$ |
| | | Lubricated | 30 MPa<br>100 MPa | $0.32 \pm 0.02$<br>$0.39 \pm 0.02$ |
| | Ground | Dry | 30 MPa<br>100 MPa | $0.33 \pm 0.05$<br>$0.42 \pm 0.02$ |
| | | Lubricated | 30 MPa<br>100 MPa | $0.28 \pm 0.02$<br>$0.43 \pm 0.03$ |
| 42CrMo4 + QT (struc.) vs. C45 | Rough turned | Dry | 30 MPa<br>100 MPa | $0.42 \pm 0.04$<br>$0.66 \pm 0.05$ |
| | | Lubricated | 30 MPa<br>100 MPa | $0.46 \pm 0.03$<br>$0.62 \pm 0.05$ |
| | Fine turned | Dry | 30 MPa<br>100 MPa | $0.30 \pm 0.01$<br>$0.50 \pm 0.13$ |
| | | Lubricated | 30 MPa<br>100 MPa | $0.32 \pm 0.02$<br>$0.56 \pm 0.05$ |
| | Ground | Dry | 30 MPa<br>100 MPa | $0.57 \pm 0.11$<br>$0.84 \pm 0.11$ |
| | | Lubricated | 30 MPa<br>100 MPa | $0.58 \pm 0.08$<br>$0.77 \pm 0.06$ |

### 3.4. Contact Surface Evaluation after Testing

In order to understand the damaging process of the microstructures, the height profiles of the contact surfaces of the specimen and counterbody were analyzed after testing. Selected surfaces are shown in Figure 11. A significantly wider tip erosion occurred for the microstructures of the 42CrMo4 + QT specimens, for both the lubricated and nonlubricated variations compared to the C45. In all variants with the higher nominal surface pressure $p_{nom} = 100$ MPa, plowing damage was noticable in the counterbody. No plowing occurred for the variation with a lower nominal surface pressure $p_{nom} = 30$ MPa. Here, an indentation of the tips of the microstructures was seen but no plowing marks. This indicates that the tips of the microstructures failed due to shearing without damaging the counterbody. These two failure mechanisms can be seen in Figures 9 and 10, where the maximum shifted towards a larger displacement at higher pressures.

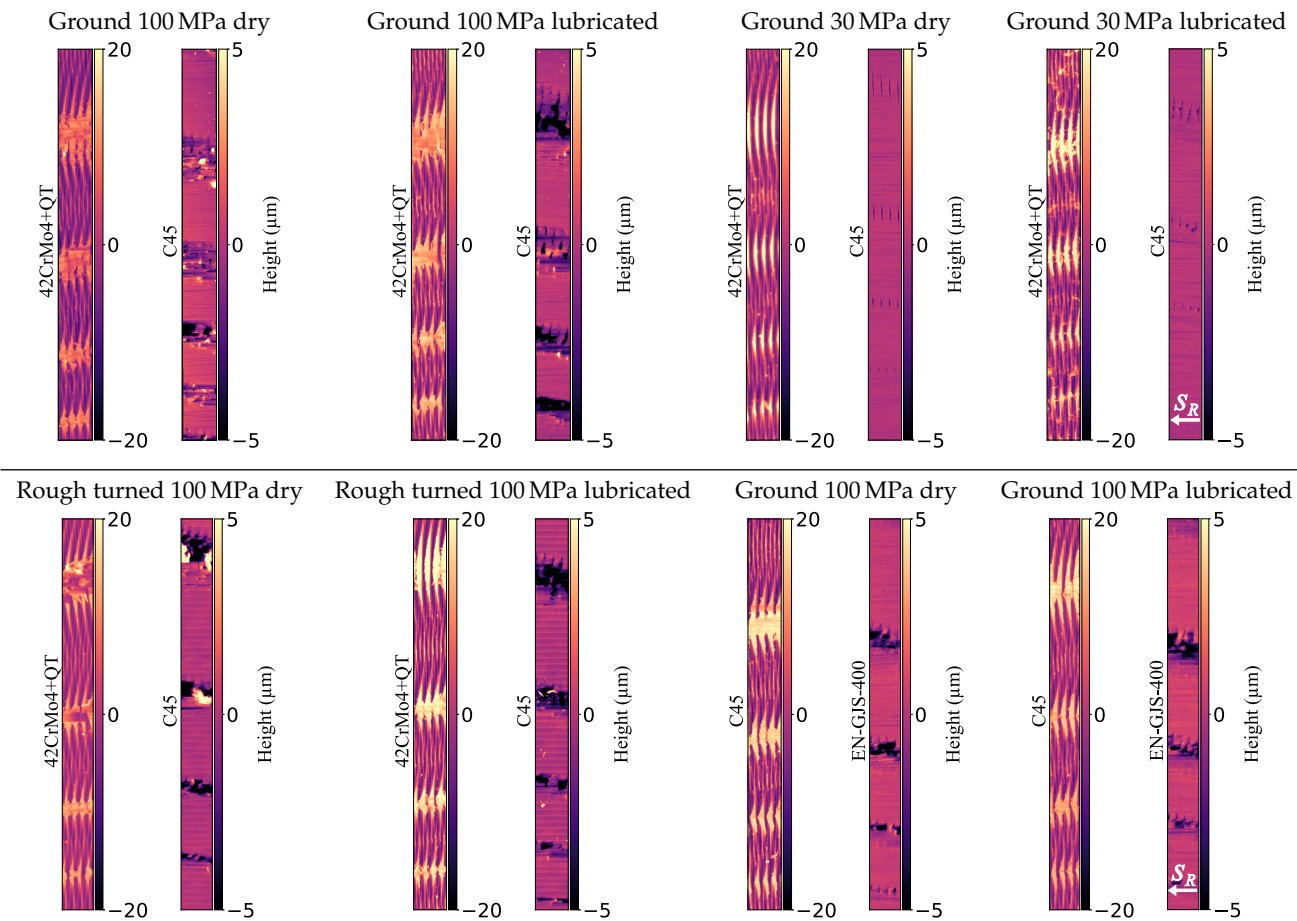

**Figure 11.** Surface height profiles of selected structured specimens and counterbodies after $s_R = 1000\,\mu m$ displacement.

After analyzing the surface height profiles of the surfaces, a deeper analysis of the corresponding surface parameters was necessary. Previous studies regarding microstructures showed a good correlation between the peak material volume $Vmp$ and $\mu_{max}$ [27]. While this parameter may provide a good estimation of the frictional behavior of different microstructures, the specimens in this study used nearly identical microstructures (Figure 3). An analysis of the correlation between the COF $\mu_{max}$ and $Vmp$ showed almost no correlation between the two parameters ($r^2 = 0.08$). Thus, an analysis of the structured surfaces did not seem useful to describe the friction behavior from the same microstructures under different tribological influences. In Figure 11, the height profiles of the counterbody surfaces showed a different behavior in the resulting damage, depending on the materials in contact and the pressure. Using this information and changing the analysis from the structured specimens to the counterbodies, a correlation between the dale void volume $Vvv$ and $\mu_{max}$ could be found. With $r^2 = 0.68$ for the tests with $p_{nom} = 100\,MPa$, these parameters showed a good correlation. This effect can be attributed to the damage to the counterbody surface caused by the microstructures. A larger resistance against shearing of the tips allowed the structures to plow further in the counter body, thus increasing the $Vvv$, while also leading to a larger transmittable torque. Another connection existed between the root mean square gradient $Sdq$ and $\mu_{max}$ with $r^2 = 0.88$. This relationship described the change in the mean surface gradient due to the damage by the structured specimen. It seems that an increase in damage also increased the mean gradient of the counterbody surface due to plowing, in a matter comparable to the dale void volume.

## 4. Conclusions

The investigation of the microstructures under different conditions showed that the microstructures generally led to an increase in the COF. Important influencing variables for the COF increase were the nominal surface pressure $p_{\text{nom}}$, the surface shape, and the materials used. The shift from adhesive to deformative friction behavior significantly reduced the influence of the surface lubrication. Building on previous studies [27], it was found that for identical structured specimens, the *Vmp* was not the only indicator for the COF. If the same microstructure is used for test specimens made of different materials, the penetration and damage behavior of the microstructures in the counterbody must be analyzed to determine the influence on the COF, rather than the microstructure itself. Two surface parameters that are suitable for describing the COF as a function of the surface structure are the dale void volume *Vvv* and the root mean square gradient *Sdq* of the counterbody. The estimation of the COF increase and the change from adhesive to deformative friction enables improving the performance of the frictional connections.

**Author Contributions:** J.S.: Investigation, Visualization, Methodology, and Writing—Original draft preparation. R.F.: Investigation, Visualization, Methodology, Writing—Original draft preparation, and Funding acquisition. A.S.: Project administration, Funding acquisition, Supervision, Review, and Methodology. A.H.: Project administration, Funding acquisition, Supervision, Review, and Methodology. All authors have read and agreed to the published version of the manuscript.

**Funding:** This work was funded by the Deutsche Forschungsgemeinschaft (DFG, German Research Foundation; grant number 411688125).

**Data Availability Statement:** The necessary data are contained within the article. Additional data are available on request.

**Conflicts of Interest:** The authors declare that they have no known competing financial interests or personal relationships that could have appeared to influence the work reported in this paper.

## Abbreviations

The following abbreviations are used in this manuscript:

COF     Coefficient of friction

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
