# Peer review of "Investigating the Friction Behavior of Turn-Milled High Friction Surface Microstructures under Different Tribological Influence Factors"

_jmmp, doi:10.3390/jmmp6060143_

Round 1
Reviewer 1 Report
The authors investigate the coefficient of friction of steel-steel and steel-cast iron pairings with and without surface microstructures. They find that surface microstructures lead to higher values of the friction coefficient, in both dry and lubricated conditions.
Their work is interesting, and the results are supported by the observations and consistent with the study design. I recommend publications after they address the minor comments that follow.
#1 On mu_adh, mu_def, and the role of lubricants:
- why do lubricants reduce mu_adh but not mu_def? what if the lubricant is pressurized or its thickness large enough to alter significantly the true contact area?
- in what conditions are their lubricated experiments run? what is the lubricant?
- how good of an estimante is mu_adh = mu_dry - mu_wet? there clearly is adhesive wear in dry conditions
#2 The authors say that microstructures penetrate in the counter-body. How does this affect the wear behaviour? Is there any correlation between wear and friction coefficient?
#3 I assume these results could change significantly if the same tests were performed at different sliding velocities. Can the authors comment on this? What applications are these tests a good proxy for?
Further comments:
line 26: I reccomend expanding on the "several mechanisms"
ref 18 and others in the text are in German thus much less accessible - are there any other equivalent references in English? if not, the others should briefly summarize/expand the point of each reference when mentioning it in the main text
l 114: where does Eq. 1 come from?
l 128 and Fig 2(right): Where does the classification in "Characteristics A/B/C" come from? It is not clear to me from the text 1) why one is more representative of one friction type than another, and 2) what is the difference between B and C
l 130: how is mu_20 determined? why 20microns? has slip started at that sliding distance?
l 202: "the third" replace with "ground"
table 2: what is the counterbody in first column? what are the unstructured and what the structured?
l 247: explain the mechanism with which embedded particles lubricate contact
Fig 8: not enough information to understand it: what conditions are the surfaces from? dry? lubricated? fine turned, ground? Fig 11 has more info.
Fig 8 and Fig 11: add more specific info about when the pics are taken (sliding distance, mu) for the post-mortem surfaces, and of the sliding direction with respect to the picture
l 334-349: interesting but unclear how the correlation is measured. Over all the tests independent of conditions? For each individual pairing?
Reviewer 2 Report
The topic of the manuscript is modern, the methods used during the research are absolutely correct, and the presentation of the results is commendable. I've reviewed several manuscripts, but this is by far the most well-crafted work I've come across. I have a few suggestions for improvement for the authors.
- It often helps to understand the formulas used if the parameters included in them are listed below them. I am thinking here, for example, of Equation 3, where the resulting torque Tr is given above it in the text, but you have to look it up in order to interpret it.
- For Figure 2, I would recommend the notation used in Figure 3 (with the letters a and b) instead of the "left" and "right" subfigure used now. This could perhaps be applied to the later figures (for example Figs. 5, 6, 7, 9 and 10).
- in Line 157 the Keyence VK-X1000 instrument is mentioned. What is the accuracy of this measurement setup?
- In lines 171-173, it is mentioned that height, hybrid and functional volume parameters were used, but the specific designations are given only one page later (From Line 215) . I have two suggestions to solve this: either the list of parameters should be moved to here, or else a sentence should be used to indicate that they will be specifically listed later.
- For the 42CrMo4 material, it is mentioned that it was in the +QT state, but for the other two materials, the material state is not defined.
- The text above Figure 5 may refer to this figure, but the figure does not show the surface image of the 42CrMo4 material mentioned in the text.
- My comment about Figure 6 arose when I tried to compare the results with Figures 9 and 10, but I had to look for the value of the surface pressure applied to the figure in the text: it would be useful to find the applied parameters in the figures: either in the figure itself or in its caption so that the text does not have to be browsed to interpret and compare the results. Back when I was still a PhD student, my professor said that everything in a diagram must be on it to be interpretable on its own. Again: this is just a suggestion, I know that sometimes it is not easy to implement.
- The sentences in lines 253 and 254 begin with the same subsentences ("This is also..."), perhaps the story could be expressed a little more eclectically.
- I would like to make a comment about the friction characteristics introduced in Figure 2: it would be advisable to highlight them in the text with bold or italic letters and mark them uniformly (e.g. Char. A, etc.), because sometimes the markings are difficult to separate from the rest of the sentences (e.g. Line 270, 294, 295).
- Finally, there is a typo in line 338: "counterbodiy".
Reviewer 3 Report
Dear Authors
I am pleased to have reviewed your work. It is a complex topic and I believe it has been adequately addressed and above all well presented.
However, there are perhaps a few comments that I would like to see indicated in the paper to improve the understanding of the results.
Line 183 talks about the chemical composition but does not indicate the method of analysis, whether by microscopy or by XRF. Since the table is only explanatory, I understand that it does not indicate patterns or uncertainty, its absence can be assumed.
In table 1 in the Mo, there is a comma, I don't know if it is a separator or a decimal.
In table 4 and in those tables where COF is discussed, the standard deviation of the results is quite low, especially in dry. Please indicate the number of tests carried out to know the validity of the results obtained.
Best regards,
